# Rice stripe virus utilizes a *Laodelphax striatellus* salivary carbonic anhydrase to facilitate plant infection by direct molecular interaction

Jing Zhao[1,2†], Xiangyi Meng[1,2†], Jie Yang[1,2], Rongxiang Fang[1,2], Yan Huo[1]*, Lili Zhang[1,3]*

[1]Department of Agri-microbiomics and Biotechnology, State Key Laboratory of Microbial Diversity and Innovative Utilization, Institute of Microbiology, Chinese Academy of Sciences, Beijing, China; [2]College of Life Sciences, University of the Chinese Academy of Sciences, Beijing, China; [3]College of Advanced Agricultural Sciences, University of the Chinese Academy of Sciences, Beijing, China

**\*For correspondence:**
huoy@im.ac.cn (YH);
zhangll@im.ac.cn (LZ)

[†]These authors contributed equally to this work.

**Competing interest:** The authors declare that no competing interests exist.

## eLife Assessment

This **valuable** study presents a well-designed set of experiments demonstrating how a planthopper salivary carbonic anhydrase can promote rice stripe virus infection by modulating callose deposition in the host plant. The authors provide **solid** data for the proposed protein–protein interactions, including strengthened evidence for the LssaCA-NP-OsTLP complex and clarified dynamics of LssaCA presence in planta. Overall, the work reveals a mechanistic link whereby a vector salivary protein enhances a plant β-1,3-glucanase to suppress callose-based defense, thereby facilitating early viral establishment.

**Abstract** Plant viruses typically rely on insect vectors for transmission between plants, with insect salivary proteins playing critical roles in this process. In this study, we demonstrate how *Laodelphax striatellus* salivary carbonic anhydrase (LssaCA) promotes rice stripe virus (RSV) infection in plants. We discovered that LssaCA directly binds to RSV nucleocapsid protein (NP) in insect salivary glands. This LssaCA-NP complex interacts with a rice thaumatin-like protein (OsTLP) that possesses endo-β-1,3-glucanase activity potentially degrading callose. Upon binding, the LssaCA-NP complex significantly enhances OsTLP enzymatic activity. We further clarify that both *L. striatellus* feeding and RSV infection induce callose deposition. The tripartite LssaCA-NP-OsTLP interaction enhances callose degradation, thereby facilitating RSV infection via its insect vector. This study provides new insights into complex virus-insect-plant tripartite interactions mediated by insect salivary proteins, with broad implications for numerous plant viruses transmitted by insect vectors.

## Introduction

During plant virus transmission by piercing-sucking insects, most viruses are inoculated into the plant phloem via the insect's secreted saliva (*Arcà and Ribeiro, 2018*; *Conway et al., 2016*; *Wu et al., 2022*). Thus, insect saliva acts as an interface for the virus–insect–host tripartite interaction and can directly promote viral transmission to, and infection of, the host plants (*Sun et al., 2020*; *Wu et al.,*

*2022*). However, despite the importance of insect salivary proteins in this tripartite interaction, there is still much to learn about how these proteins enable successful viral infection.

Previous studies have revealed that there are two ways in which insect saliva facilitates viral infection. One is an indirect approach whereby the saliva modulates the host microenvironment at the feeding site, and saliva effectors work together to allow the arthropod to go unnoticed while it feeds on the host plant (*Acevedo et al., 2019*; *Arcà and Ribeiro, 2018*; *Sun et al., 2020*). For example, our work indicated that a *Laodelphax striatellus* mucin protein, LssaMP, enables the formation of the salivary sheath and facilitates the transmission of rice stripe virus (RSV) into the rice phloem (*Huo et al., 2022*). The study on leafhoppers revealed that the expression of a saliva calcium-binding protein is inhibited by rice gall dwarf virus (RGDV), thus causing an increase of cytosolic $Ca^{2+}$ levels in rice and triggering callose deposition and $H_2O_2$ production. This increases the frequency of insect probing, thereby enhancing viral horizontal transmission into the rice phloem (*Wu et al., 2022*). The other mechanism by insect saliva to facilitate virus infection is direct regulation, whereby saliva proteins promote virus transmission through specific molecular interactions (*Wen et al., 2019*). Direct saliva protein–pathogen interactions have been reported in animal pathogens. For example, during transmission of *Borrelia burgdorferi* by *Ixodes scapularis*, the saliva protein Salp15 of *I. scapularis* binds to the bacterial outer surface protein C, which prevents the bacterium from being recognized by the animal immune system. In this way, the saliva protein enables the pathogen to infect the animal host (*Ramamoorthi et al., 2005*; *Schuijt et al., 2008*). Although most plant viruses are heavily dependent on insect vectors for plant-to-plant transmission (*Gray, 2008*), the direct function of insect saliva proteins in mediating virus transmission remains largely uninvestigated.

During sap-feeding, arthropods produce two distinct types of saliva at different stages of the feeding process: gel saliva and watery saliva (*Bonaventure, 2012*; *Lou et al., 2019*). The former forms a salivary sheath to provide a smooth path for the stylet penetration (*Lou et al., 2019*). The latter is mainly secreted into the phloem sieve elements to prevent them from plugging up and suppress plant defense responses. Some salivary components may act as herbivore-associated molecular patterns that can trigger pattern-triggered immunity, and certain salivary effectors may be recognized by plant resistance proteins to induce effector-triggered immunity, etc. (*Huang et al., 2017*; *Ji et al., 2017*; *Yi et al., 2021*).

For phloem-feeding insects, callose deposited on phloem sieve plate and plasmodesmata of sieve elements functions as a defense mechanism by reducing insect feeding and preventing viral movement (*Hao et al., 2008*; *Hipper et al., 2013*; *Will and Vilcinskas, 2015*; *Zavaliev et al., 2011*; *Yue et al., 2022*). Callose is a β-(1,3)-D-glucan polysaccharide that is synthesized by callose synthases and degraded by β-(1,3)-glucanases. Plants defend themselves by depositing callose at the sieve plates and plasmodesmata in response to virus infection, whereas viruses counter this defense by activating β-(1,3)-glucanases to degrade callose (*Bucher et al., 2001*; *Hao et al., 2008*; *Wu et al., 2022*; *Zavaliev et al., 2011*).

RSV is the causative agent of rice stripe disease, a serious disease of rice crops that has occurred repeatedly in China, Japan, and Korea (*Xu et al., 2021*). RSV is completely dependent on insect vectors for transmission among its host plants, and *L. striatellus* is the main vector (*Xu et al., 2021*; *Zhao et al., 2017*). *L. striatellus* transmits RSV in a persistent-propagative manner. The virus initially infects the midgut, then disperses from the hemolymph into the salivary glands and is inoculated into the plant host during *L. striatellus* feeding (*Huo et al., 2022*). *L. striatellus* belongs to the order Hemiptera, whose members mainly feed from sieve tubes through their mouthparts (stylets) that penetrate plant tissues and reach sieve tubes to ingest the phloem sap (*Tjallingii, 2006*; *van Bel and Will, 2016*). RSV is mainly secreted into the rice phloem via the watery saliva (*Huo et al., 2022*; *Wang and Blanc, 2021*).

In this study, we identified a molecular interaction among RSV, an *L. striatellus* saliva protein, and a plant β-1,3-glucanase. The insect saliva protein directly binds to the RSV nucleocapsid protein (NP) and then binds to a rice thaumatin-like protein to activate its β-1,3-glucanase activity. The activation of β-1,3-glucanase helps RSV infection by inhibiting callose deposition in response to viral infection.

## Results

### *L. striatellus* salivary carbonic anhydrase directly binds to RSV NP

Insect-secreted watery saliva is inoculated into plant phloem sieve tubes along with RSV (*Huo et al., 2022*). We identified the salivary proteins that directly interact with RSV to promote viral plant infection. Among the saliva proteins identified in our previous study from RSV-free *L. striatellus* (*Huo et al., 2022*), Sap6 (GenBank accession no. RZF48846.1) was found to co-localize with RSV in infected salivary glands, suggesting a potential molecular interaction between them (*Figure 1A*, *Figure 1—figure supplement 1*).

Sequence homology analysis revealed Sap6 as a carbonic anhydrase-like protein, with a signal peptide located at the N-terminus (amino acids 1–24) (*Figure 1B*, *Figure 1—figure supplement 2A*). We recombinantly expressed this protein in *Escherichia coli* or in sf9 cells as a secreted protein. The purified proteins were measured for enzymatic activity in hydrolyzing *p*-nitroethyl acetate. The observed hydrolyzing activity confirmed Sap6 as a carbonic anhydrase (abbreviated as CA, *Figure 1C*). Sequence analysis indicated seven residues constituting the enzymatic active center: $H_{111}$, $N_{139}$, $H_{141}$, $H_{143}$, $E_{153}$, $H_{166}$, and $T_{253}$. We then constructed a mutant with substitutions of all seven residues (*Supplementary file 1*. H111D, N139H, H141D, H143D, E153H, H166D, and T253E). The mutant protein lost enzymatic activity, indicating the necessity of the enzymatic active center (*Figure 1C*). Sap6 was subsequently named *L. striatellus* salivary carbonic anhydrase (LssaCA).

We performed pull-down and microscale thermophoresis (MST) assays to confirm the molecular interaction between LssaCA and RSV NP. Pull-down assays revealed a specific interaction between LssaCA and NP (*Figure 1D*). MST assays determined that the $K_D$ of the interaction between LssaCA and NP was 2.7±2.2 µM, further confirming the molecular interaction (*Figure 1E*, *Figure 1—figure supplement 3*).

We then performed quantitative reverse-transcriptional PCR (RT-qPCR) to determine the gene expression and western blotting to assess protein distribution. RT-qPCR revealed specific expression of *LssaCA* in *L. striatellus* salivary glands (*Figure 1F*), and western blotting confirmed the protein's presence in this tissue (*Figure 1F*). When *LssaCA* expression was knocked down using *LssaCA*-specific double-stranded RNA (dsLssaCA) microinjection, LssaCA protein levels in salivary glands decreased dramatically (*Figure 1G*). To investigate LssaCA secretion, we allowed RSV-free insects to feed on an artificial diet, collecting watery saliva proteins from the diet solution and gel saliva proteins from the feeding parafilm. Western blotting demonstrated that the LssaCA protein was secreted as a component of watery saliva (*Figure 1H*). Furthermore, by measuring LssaCA protein levels in plants immediately after feeding, we confirmed that LssaCA can be secreted into plants (*Figure 1I*). To quantify LssaCA protein levels following insect feeding, we allowed 50 RSV-free insects to feed on a confined plant area for 2 days. Western blotting revealed that LssaCA persisted for at least 3 days post-feeding (dpf), with protein levels declining rapidly between 1–3 dpf (*Figure 1J*). We also examined the regulation of *LssaCA* expression by RSV infection. While RSV infection did not significantly alter *LssaCA* mRNA levels in the salivary glands, the protein levels were significantly elevated in RSV-infected salivary glands (*Figure 1K*).

In conclusion, these results indicate that when RSV NP is secreted by *L. striatellus*, a salivary-gland-specifically expressed protein, LssaCA, can directly bind to RSV and be secreted via saliva.

### LssaCA enhances RSV infection in plants

To determine the role of LssaCA in RSV infection of rice plants, 36.8 nL dsLssaCA at 1 ng/nL was microinjected into the hemocoel of RSV-infected third-instar nymphs to interfere with gene expression. At 2 days post-microinjection (dpi), *LssaCA* mRNA levels were measured by RT-qPCR and compared with the dsGFP control. Results showed that dsLssaCA treatment reduced *LssaCA* mRNA levels by 80% (*Figure 2A*). The two groups of insects were then allowed to feed on healthy rice seedlings (five insects per rice seedling) for 2 days. RSV NP RNA levels (indicative of RSV titers) in the plants were measured at 14 dpf, when inoculated viruses had infected the plant host, replicated in plant cells, and viral titers were high enough to be detected by RT-qPCR and western blotting. Compared to rice seedlings fed by dsGFP-treated insects, those fed by dsLssaCA-treated insects had significantly reduced RSV titers (*Figure 2B and C*), indicating that LssaCA plays an essential role in mediating RSV infection of rice plants.

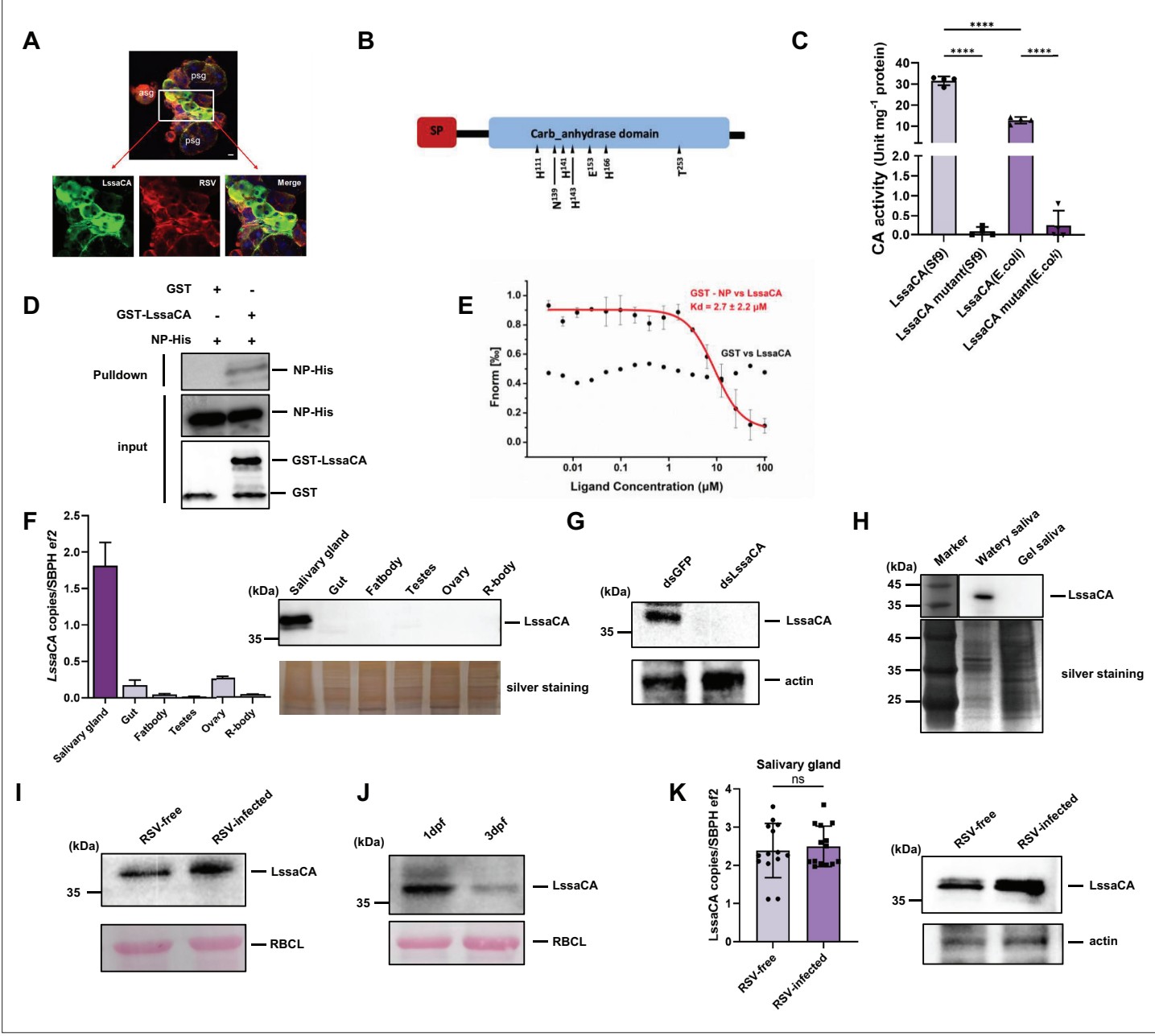

**Figure 1.** An *L. striatellus* salivary carbonic anhydrase (LssaCA) directly binds to rice stripe virus (RSV) nucleocapsid protein (NP) in salivary glands. (**A**) Immunofluorescence showing co-localization of RSV (red) and LssaCA (green) in *L. striatellus* salivary glands. Images represent three independent experiments with 30 SBPHs analyzed. Scale bar: 20 μm. psg, principal salivary gland; asg, accessory salivary gland. (**B**) Schematic of LssaCA protein sequence. SP, signal peptide; Carb_anhydrase domain, conserved eukaryotic-type carbonic anhydrase sequence. Triangles indicate seven predicted catalytically active residues. (**C**) Esterase activity of recombinant LssaCA protein expressed in sf9 insect cell and *E. coli* system. In LssaCA mutant protein, all seven predicted catalytically active residues were replaced. Mean and SD were calculated from 2 independent enzymatic assays. ****p<0.0001. (**D**) GST pull-down assays demonstrating interaction between LssaCA (GST-LssaCA) and RSV NP (NP-His). GST served as negative control. Anti-GST and anti-His antibodies detected corresponding proteins. (**E**) MST assay showing binding between LssaCA (LssaCA-His) and RSV NP (GST-NP). GST served as negative control. Error bars represent SD. (**F**) Measurement of *LssaCA* tissue-specific expression by RT-qPCR and western blotting. Third-instar RSV-free nymphs were used to measure the gene expression in gut, salivary gland, fat body, and remaining tissues (R-body) after dissection of the aforementioned tissues. Adult females and adult males were used to measure gene expression in ovaries and testes, respectively. Mean and SD were calculated from three independent experiments with 5 tissue samples each experiment. LssaCA protein was detected using anti-LssaCA antibodies. (**G**) Western blot analysis of LssaCA protein in RSV-free SBPH salivary glands. dsLssaCA treatment significantly reduces target protein levels compared to controls. LssaCA protein was detected using anti-LssaCA antibodies. (**H**) Western blot analysis showing LssaCA distribution in watery and gel saliva of RSV-free insects. LssaCA protein was detected using anti-LssaCA antibodies. Total saliva proteins were visualized by silver staining. (**I**) Western blot

*Figure 1 continued on next page*

*Figure 1 continued*

detection of LssaCA proteins in rice phloem immediately following *L. striatellus* feeding. (**J**) Western blot detection of LssaCA proteins in rice phloem at 1 and 3 days post-feeding (dpf). (**I, J**). LssaCA protein was detected using anti-LssaCA antibodies. Rubisco large subunit (RBCL) served as protein loading control. (**K**) Analyses of LssaCA levels regulated by RSV infection. Third-instar RSV-free and RSV-infected nymphs were used to measure gene expression (RT-qPCR) and protein accumulation (Western blotting). Mean and SD were calculated from three independent experiments with a total of 14 tissue samples each experiment. ns, not significant. All western blot images are representative of three independent experiments.

The online version of this article includes the following source data and figure supplement(s) for figure 1:

**Source data 1.** Original files for for the Western blot, silver staining, and Ponceau S staining in *Figure 1*.

**Source data 2.** PDF file containing original western blots silver staining, and Ponceau S staining for *Figure 1*, indicating the treatments.

**Figure supplement 1.** Immunofluorescence localization of rice stripe virus (RSV) and *L. striatellus* salivary carbonic anhydrase (LssaCA) in uninfected *L. striatellus* salivary glands.

**Figure supplement 2.** Characteristics of LssaCA.

**Figure supplement 3.** SDS-PAGE analysis of purified recombinant proteins used in microscale thermophoresis (MST) assays.

**Figure supplement 3—source data 1.** Original files for for Coomassie-stained gels in *Figure 1—figure supplement 3*.

**Figure supplement 3—source data 2.** PDF file containing original image of Coomassie-stained gels for *Figure 1—figure supplement 3*, indicating the treatments.

To determine at which step LssaCA affected RSV infection, we analyzed the influence of LssaCA deficiency on RSV titers during several steps of the RSV transmission and infection process, including virus load in salivary glands and saliva, and initial inoculation levels in planta. First, at 2 days after dsLssaCA treatment of viruliferous insects, when *LssaCA* was significantly downregulated (*Figure 2D*), RT-qPCR was used to detect RSV titer in insect salivary glands. Results showed that *LssaCA* deficiency did not reduce RSV titers in this tissue (*Figure 2E*). Second, LssaCA-deficient insects were allowed to feed on an artificial diet, and their saliva was collected for RSV level analysis. Results revealed that LssaCA deficiency did not reduce RSV titer in the secreted saliva (*Figure 2F*). Third, a rice plant was fed with 10 viruliferous insects for 24 h, with insects limited to a specific area, and RSV titers were measured immediately after feeding. We found that LssaCA deficiency did not affect RSV initial inoculation levels in planta (*Figure 2G*). Fourth, we measured RSV infection levels at 3 dpf to assess the in-planta influence after viral inoculation. *LssaCA* expression was knocked down using dsLssaCA in RSV-infected third-instar nymphs, and five insects were allowed to feed on healthy rice plants for 2 days. We then measured RSV titers in whole plant shoots at 3 dpf. Significantly lower RSV titers were observed in the dsLssaCA-treatment group at 3 dpf (*Figure 2H and I*).

Collectively, these results suggest that LssaCA promotes RSV infection through a mechanism occurring not in insects or during early stages of viral entry in plants, but in planta after viral inoculation.

## LssaCA interacts with rice thaumatin-like protein to increase its endo-β-1,3-glucanase activity

To investigate the mechanism by which LssaCA facilitates RSV infection in plants, we performed yeast two-hybrid screening to identify rice proteins that interact with LssaCA. A rice thaumatin-like protein (XP_025879846.1), designated as *Oryza sativa* thaumatin-like protein (OsTLP) (*Figure 3A and B*, *Figure 3—figure supplement 1*, and *Supplementary file 2*), was identified. The open reading frame of OsTLP encodes a polypeptide of 327 amino acids with a thaumatin family (THM) domain from residues 27–267. A BLAST algorithm-based analysis revealed sequence identities to TLP 1b. The N-terminal 24 amino acids constitute a secretion signal peptide, suggesting OsTLP is a secreted protein. Pull-down and MST assays were performed to confirm the molecular interaction between OsTLP and LssaCA. Pull-down assays showed that OsTLP and LssaCA have a specific interaction (*Figure 3C*). The MST assay revealed a $K_D$ of 1.3±2.2 µM for the OsTLP-LssaCA interaction, further confirming the molecular interaction (*Figure 3D*).

Previous studies have shown that TLP orthologs in various plants, including apples, cherries, tomatoes, and barley, possess endo-β-1,3-glucanase activity that degrades callose (*de Jesús-Pires et al., 2020*). To verify the endo-β-1,3-glucanase activity of OsTLP, we conducted enzymatic assays using recombinantly expressed OsTLP protein. The purified protein exhibited an average activity of 60 units/mg (*Figure 3E*). To examine OsTLP activity in planta, we constructed transgenic plants overexpressing OsTLP (OsTLP OE) by using the *Japonica* rice cultivar Nipponbare (*Figure 3—figure supplement 2A*

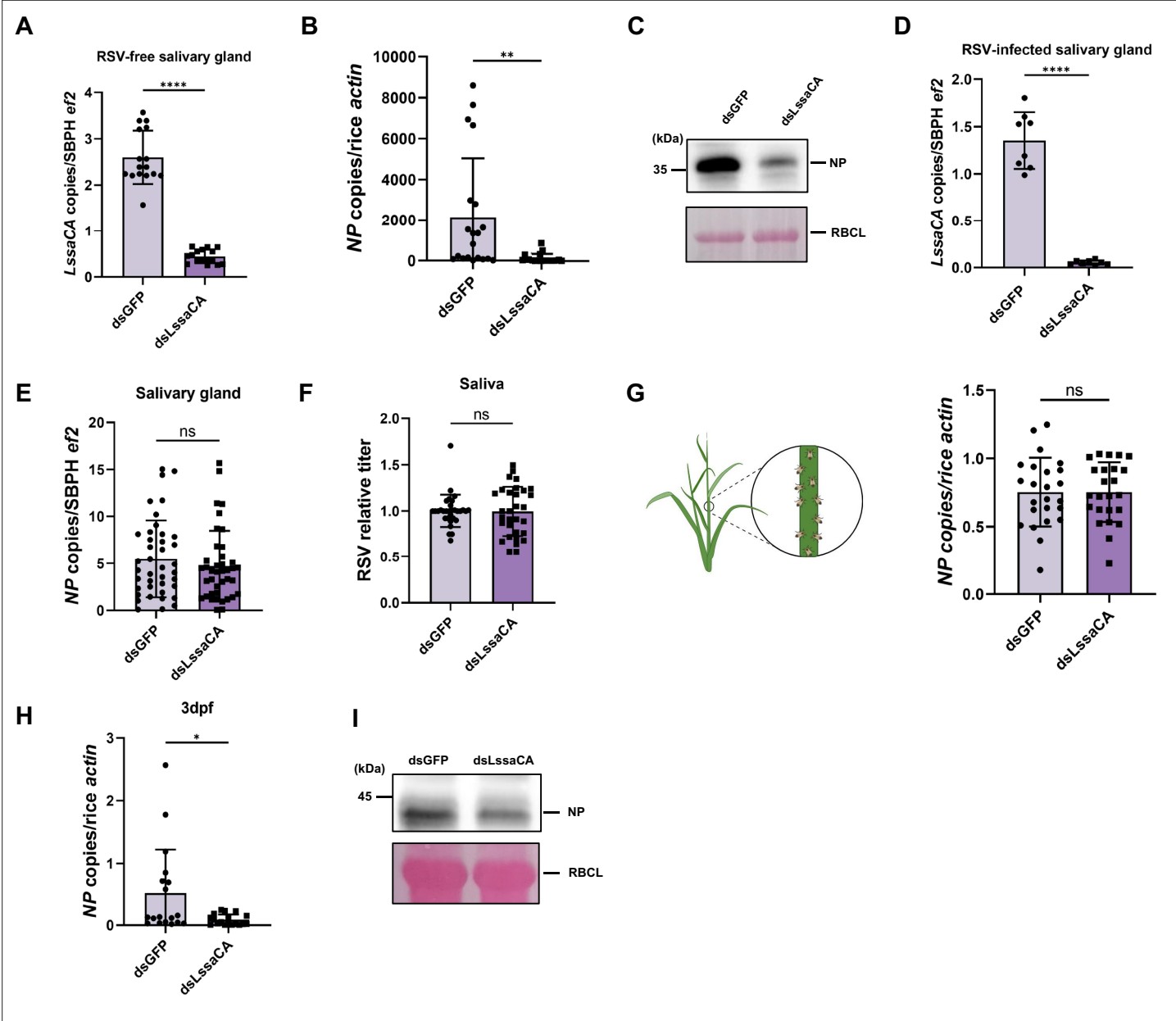

**Figure 2.** *L. striatellus* salivary carbonic anhydrase (LssaCA) enhances rice stripe virus (RSV) infection in rice plants. (**A**) Gene silencing efficiency determined by RT-qPCR. (**B, C**) Analysis of RSV infection levels in rice seedlings. Each plant seedling was fed upon by five infected insects for 2 days, followed by 14 days of cultivation. RSV levels were determined by RT-qPCR (**B**) and western blotting (**C**). RSV NP detected by anti-NP polyclonal antibodies indicated viral titers. Rubisco large subunit (RBCL) served as protein loading control. (**D**) Gene silencing efficiency determined by RT-qPCR. (**E**) RT-qPCR analysis of RSV titers in salivary glands. Each dot represents salivary glands from five RSV-infected third-instar nymphs. (**F**) RSV titers in *L. striatellus* saliva determined by RT-qPCR. Each dot represents one saliva sample from 10 insects fed with artificial diet. (**G**) RT-qPCR analysis of RSV titers in rice. Plants were fed upon by infected insects for 24 h (left: schematic of *L. striatellus* feeding on specific rice stem sites), with titers assayed immediately post-feeding (right). Each dot represents an individual rice seedling. (**H, I**) RSV infection in plants at 3 dpf. Five RSV-infected third-instar *L. striatellus* nymphs were fed upon healthy rice plants for 2 days, and viral titers in whole plant shoots were measured at 3 dpf by RT-qPCR (**H**) and western blotting (**I**). Each dot represents an individual plant. RSV NP detected by anti-NP polyclonal antibodies indicated viral titers. RBCL protein served as a protein loading control. Mean and SD were calculated from three independent microinjection experiments. ns, not significant; *p<0.05; **p<0.01; ****p<0.0001. All western blot images are representative of three independent experiments.

The online version of this article includes the following source data for figure 2:

**Source data 1.** Original files for for the Western blot and Ponceau S staining in *Figure 2*.

**Source data 2.** PDF file containing original western blots and Ponceau S staining for *Figure 1*, indicating the treatments.

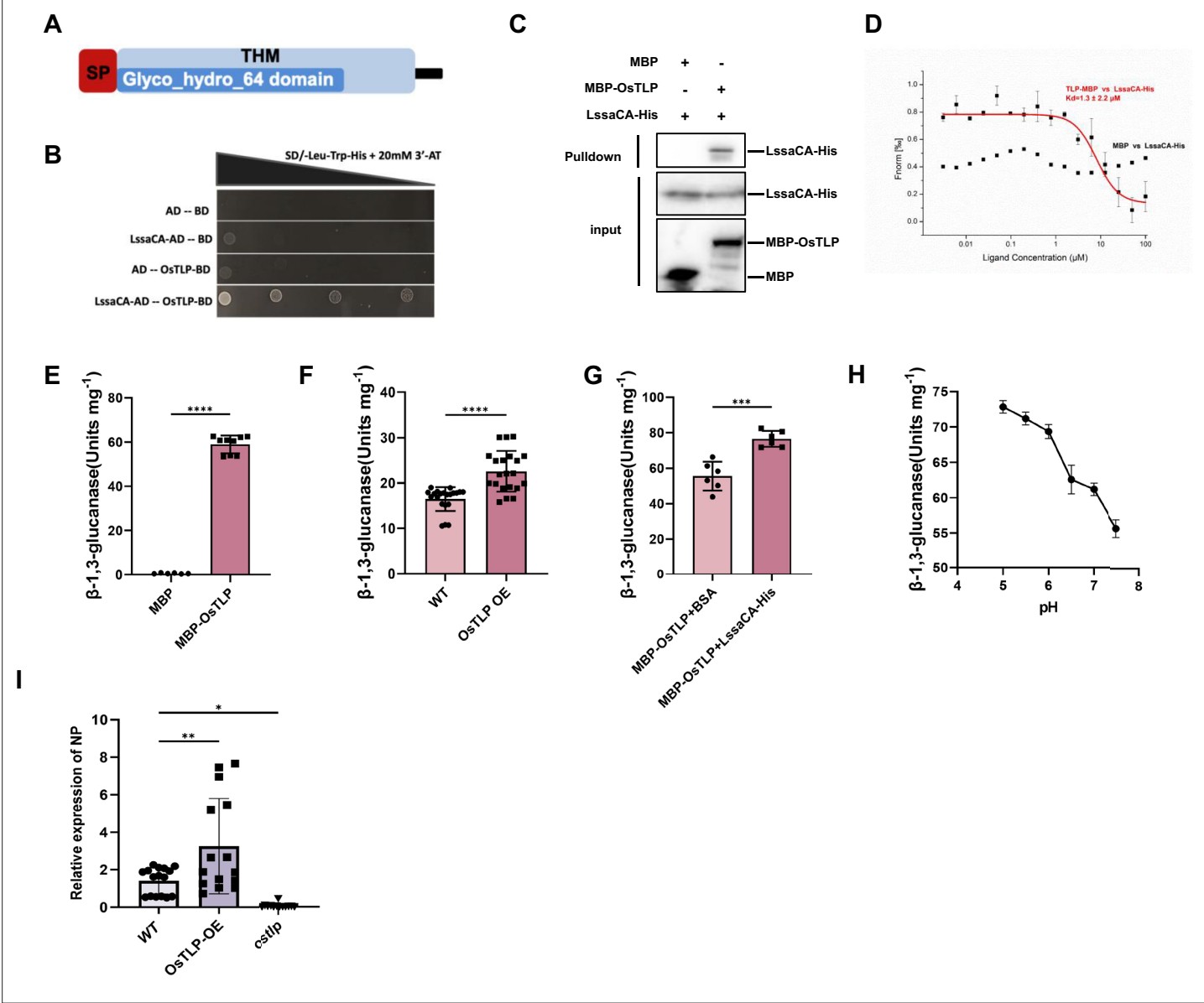

**Figure 3.** *L. striatellus* salivary carbonic anhydrase (LssaCA) interacts with rice thaumatin-like protein (OsTLP) to enhance its endo-β-1,3-glucanase activity. (**A**) Schematic diagram of OsTLP protein sequence. SP, signal peptide; THM, conserved thaumatin family protein sequence. Glyco_hydro_64 domain, glycoside hydrolases of family 64. (**B**) Yeast two-hybrid assay showing interaction between OsTLP and LssaCA. SD, synthetically defined medium; Leu, Leucine; Trp, Tryptophan; His, Histidine; 3'AT, 3-amino-1,2,4-triazole; AD, transcription activation domain; BD, DNA-binding domain. (**C**) Pull-down assays showing interactions between LssaCA (LssaCA-His) and OsTLP (MBP-OsTLP). MBP served as a negative control. Anti-His and anti-MBP polyclonal antibodies were used for protein detection. Western blot images are representative of three independent experiments. (**D**) MST assay showing molecular interactions between LssaCA (LssaCA-His) and OsTLP (MBP-OsTLP). MBP served as a negative control. Error bars represent SD. (**E**) Endo-β-1,3-glucanase activity of purified recombinant OsTLP protein expressed as MBP-fusion protein. MBP was used for normalization. (**F**) Endo-β-1,3-glucanase activity in OsTLP-overexpressing transgenic plants (OsTLP OE) or wild-type plants (WT). Plant total soluble proteins were used for enzymatic activity assays. Units mg$^{-1}$ mean units per mg of plant total soluble proteins. (**G**) Regulation of OsTLP enzymatic activity by LssaCA binding. (**H**) OsTLP enzymatic activity measured at pH 5.0, 5.5, 6.0, 6.5, 7.0, and 7.5 to assess pH dependence. (**I**) RT-qPCR analysis of RSV infection in OsTLP OE, *ostlp* mutant, and WT plants. RSV titers (NP copy number) were measured at 14 dpf. ns, not significant; *p<0.05; **p<0.01; ***p<0.001; ****p<0.0001.

The online version of this article includes the following source data and figure supplement(s) for figure 3:

**Source data 1.** Original files for Western blot analysis shown *Figure 3*.

**Source data 2.** PDF file containing original images for the Western blot for in *Figure 3*, indicating the treatments.

**Figure supplement 1.** Nucleotide and deduced amino acid sequence of OsTLP.

*Figure 3 continued on next page*

*Figure 3 continued*

**Figure supplement 2.** Generation of OsTLP-overexpressing and knockout rice lines.

**Figure supplement 2—source data 1.** Original files for Western blot analysis and Ponceau S staining in *Figure 3—figure supplement 2*.

**Figure supplement 2—source data 2.** PDF file containing original images of Western blot and Ponceau S staining for *Figure 3—figure supplement 2*, indicating the treatments.

*and B*). These transgenic plants showed significantly increased β-1,3-glucanase activity compared to wild-type controls (*Figure 3F*).

To determine whether the interaction between LssaCA and OsTLP affects enzymatic activity, we pre-incubated OsTLP with either LssaCA or BSA (control) for 2 h before measuring activity. Pre-incubation with LssaCA significantly enhanced OsTLP activity compared to the BSA control (*Figure 3G*). Since CA family proteins catalyze the conversion of $CO_2$ and $H_2O$ to $HCO_3^-$ and $H^+$, potentially lowering pH, we examined the pH dependence of OsTLP activity. Activity assays conducted across a pH range of 5.0–7.5 revealed that OsTLP activity increased at lower pH values (*Figure 3H*). However, pH monitoring during OsTLP-LssaCA interaction experiments showed no change from the initial pH of 7.4, indicating that the enhanced enzymatic activity was not due to pH changes but rather to the direct interaction between OsTLP and LssaCA. Collectively, these results suggest that LssaCA may promote RSV infection by enhancing OsTLP's callose-degrading activity.

## OsTLP facilitates RSV infection in planta

To determine the role of OsTLP in RSV infection of rice plants, we constructed *ostlp* mutant plants (*Figure 3—figure supplement 2C* and *Supplementary file 3*). Viruliferous *L. striatellus* were allowed to feed on OsTLP OE plants, *ostlp* mutant plants, or wild-type plants for 2 days, and RSV titers were measured at 14 dpf. RT-qPCR analyses indicated that compared with wild-type plants, plants

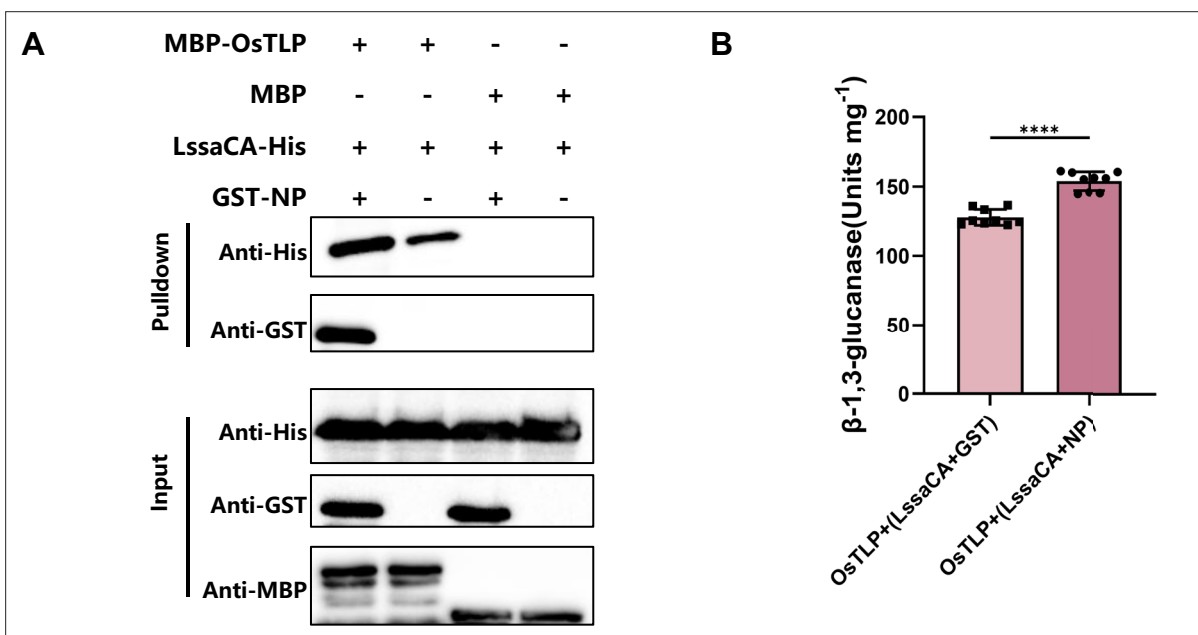

**Figure 4.** Rice stripe virus (RSV) NP-LssaCA-OsTLP tripartite interaction further enhances OsTLP enzymatic activity. (**A**) Pull-down assays showing interactions between OsTLP (MBP-OsTLP) and LssaCA-NP complex. LssaCA was expressed with a His tag, and RSV NP was expressed as a GST-fusion protein. LssaCA and NP were pre-incubated before co-incubation with OsTLP. Anti-His, anti-GST, and anti-MBP antibodies were used to detect corresponding proteins. (**B**) Endo-β-1,3-glucanase activity of OsTLP after co-incubation with LssaCA (LssaCA + GST) and LssaCA-NP complex (LssaCA + NP). LssaCA was pre-incubated with either GST or NP before addition to OsTLP. ****p<0.0001.

The online version of this article includes the following source data for figure 4:

**Source data 1.** Original files for Western blot analysis in *Figure 4*.

**Source data 2.** PDF file containing original image of Western blot for *Figure 4*, indicating the treatments.

overexpressing OsTLP had significantly higher RSV titers, while *ostlp* mutant plants exhibited significantly lower RSV titers (*Figure 3I*). These results indicate that OsTLP facilitates RSV infection in planta.

## RSV NP-LssaCA-OsTLP tripartite interaction further enhances OsTLP enzymatic activity

Given that LssaCA interacts with both RSV NP and OsTLP, we investigated the possibility of a tripartite interaction. Through pull-down assays, we discovered that when LssaCA was pre-incubated with NP to form an NP-LssaCA complex, this complex retained the ability to bind to OsTLP (*Figure 4A*). This result demonstrates a tripartite interaction among these three proteins.

We then explored how the tripartite interaction affects OsTLP enzymatic activity. OsTLP was pre-incubated with either the LssaCA-NP complex or LssaCA alone as a control. After 2 h of incubation, enzyme activity was measured. The results showed that pre-incubation with the LssaCA-NP complex significantly increased OsTLP activity compared to the LssaCA control (*Figure 4B*). This indicates that stronger activation of β-1,3-glucanase activity is achieved through the NP-LssaCA-OsTLP tripartite interaction.

## LssaCA inhibits insect-feeding and RSV-infection induced callose deposition in rice plants

Callose deposition on sieve plates and plasmodesmata serves as a critical defense mechanism in plants, reducing insect feeding, viral spread, and infection severity (*Zavaliev et al., 2011*). Given that the LssaCA or LssaCA-NP complex enhances OsTLP's β-1,3-glucanase activity and potentially promotes callose degradation, we examined callose regulation during feeding by both RSV-free and RSV-infected *L. striatellus*. To assess feeding-induced callose deposition, we exposed rice seedlings to either RSV-free or RSV-infected *L. striatellus* for 24 h and analyzed callose levels around the feeding sites. ELISA showed that while feeding by RSV-free insects increased callose concentration, feeding by RSV-infected insects triggered an even greater increase (*Figure 5A*). Consistent with these findings, microscopic analysis revealed enhanced callose deposition in phloem cells of plants fed upon by RSV-infected *L. striatellus* compared to those fed by RSV-free insects (*Figure 5B and C*, *Figure 5—figure supplement 1*). To determine whether this response was localized, we measured callose levels in tissues distant from the feeding sites. No significant differences in callose deposition were observed between control plants and the non-feeding areas (*Figure 5—figure supplement 2A*), indicating that callose accumulation was confined to feeding sites and their immediate vicinity.

To evaluate the molecular mechanisms underlying the induced callose deposition, we analyzed transcript levels of callose synthase genes using RT-qPCR. In agreement with the callose deposition patterns, feeding by RSV-free insects evaluated the expression of callose synthase genes, including *gsl3*, *gsl4*, *gsl5*, and *gsl10*, while RSV infection further enhanced their upregulation (*Figure 5D–G*, *Figure 5—figure supplement 2B–E*). These results further support that both insect feeding and RSV infection promote callose deposition. Notably, the transcriptional levels of these genes remained unchanged in non-feeding areas (*Figure 5—figure supplement 2B–E*). We also measured the *OsTLP* gene expression and found that neither SBPH feeding nor RSV infection significantly induced *OsTLP* expression (*Figure 5H*).

To investigate whether LssaCA influences callose deposition induced by insect-transmitted RSV, we downregulated *LssaCA* in RSV-infected *L. striatellus* using dsRNA microinjection. At 2 dpi, when *LssaCA* expression was significantly reduced, we allowed the insects to feed on healthy plants for 24 h before measuring callose deposition. Our results showed that plants fed upon by *LssaCA*-deficient insects accumulated more callose compared to those fed upon by control insects (dsGFP treatment) (*Figure 5I–K*). These results indicate that LssaCA inhibits callose deposition.

Using electrical penetration graph (EPG) analysis, we monitored the influence of LssaCA deficiency on the insect feeding behaviors. EPG waveforms can be classified into three different periods: nonpenetration (NP), penetration (N1, N2, N3), and phloem feeding (N4: N4-a, N4-b) periods. The NP waveform indicates a lack of feeding, the N waveform indicates penetration of mesophyll cells to seek the phloem, and the N4 waveform indicates digestion of the sap from the phloem. Results indicated that LssaCA deficiency did not affect the time to first non-phloem feeding, total feeding time in the phloem, and the time to first phloem feeding within the 24-h-detection period (*Figure 5—figure supplement 3A–D*). However, the continuity of sap ingestion was disturbed—the N4 waveform of

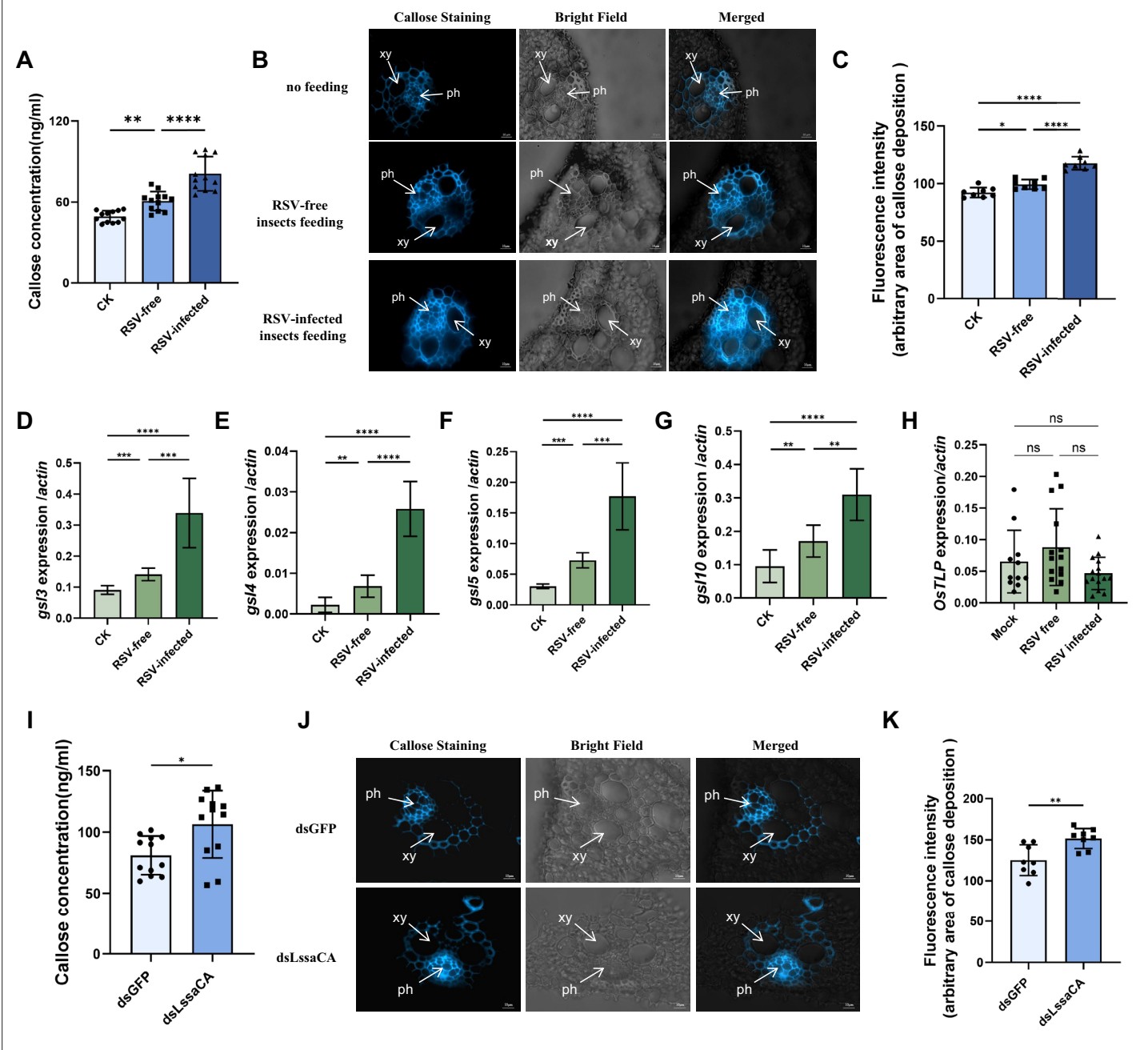

**Figure 5.** *L. striatellus* salivary carbonic anhydrase (LssaCA) inhibits *L. striatellus*-transmitted rice stripe virus (RSV)-induced callose deposition in rice plants. (**A**) Callose concentration quantified by ELISA in leaves of rice plants fed on by RSV-free or RSV-infected *L. striatellus*. Non-fed plants served as the control (CK). (**B**) Bright blue fluorescence of cross-sections showing callose deposition at feeding sites. Samples were prepared from the leaf phloem of plants that were not fed or were fed on by RSV-free or RSV-infected *L. striatellus*. Thin sections were stained with 0.1% aniline blue at 24 h post-feeding and examined under a fluorescence microscope. xy, xylem; ph, phloem. Scale bars: 10 µm. (**C**) Average fluorescence intensity of callose deposition measured using ImageJ. Eight to ten random sites per sample were selected for intensity quantification. (**D–H**) Transcript levels of callose synthase genes (*gls3, gls4, gls5, gls10*) and *OsTLP* determined by RT-qPCR. Insects were allowed to feed on rice plants for 24 h prior to RNA extraction. (**I**) Callose concentration quantified by ELISA in leaves of rice plants fed on by dsGFP- or dsLssaCA-treated RSV-infected *L. striatellus*. (**J**) Bright blue fluorescence of cross-sections showing callose deposition at feeding sites. Samples prepared from the leaf phloem of plants fed on by dsGFP- or dsLssaCA-treated RSV-infected *L. striatellus*. Scale bars: 20 µm. (**K**) Average fluorescence intensity of callose deposition measured using ImageJ. Eight to ten random sites per sample were selected for intensity quantification. Mean and SD were calculated from three independent experiments. ns, not significant; *p<0.05; **p<0.01; ***p<0.001; ****p<0.0001.

The online version of this article includes the following figure supplement(s) for figure 5:

*Figure 5 continued on next page*

*Figure 5 continued*

**Figure supplement 1.** Detection of callose deposition following *L. striatellus* feeding.

**Figure supplement 2.** Callose deposition analysis in non-feeding areas.

**Figure supplement 3.** Electrical penetration graph (EPG) to show *L. striatellus* salivary carbonic anhydrase (LssaCA) effects on *L. striatellus* feeding behaviors.

dsLssaCA SBPHs was occasionally interrupted for brief periods (*Figure 5—figure supplement 3E*). It is predicated that phloem blockage might be a reason for this feeding disruption.

In conclusion, these results demonstrate that both *L. striatellus* feeding and RSV infection induce callose deposition, which can be suppressed by LssaCA.

## LssaCA mediates in planta callose degradation by enhancing OsTLP enzymatic activity

*Ostlp* mutant plants were used to confirm whether OsTLP degrades the deposited callose induced by insect-transmitted RSV in planta. Both ELISA and microscopic analysis showed that compared to wild-type plants, *ostlp* plants exhibited significantly higher callose levels (*Figure 6A and B*).

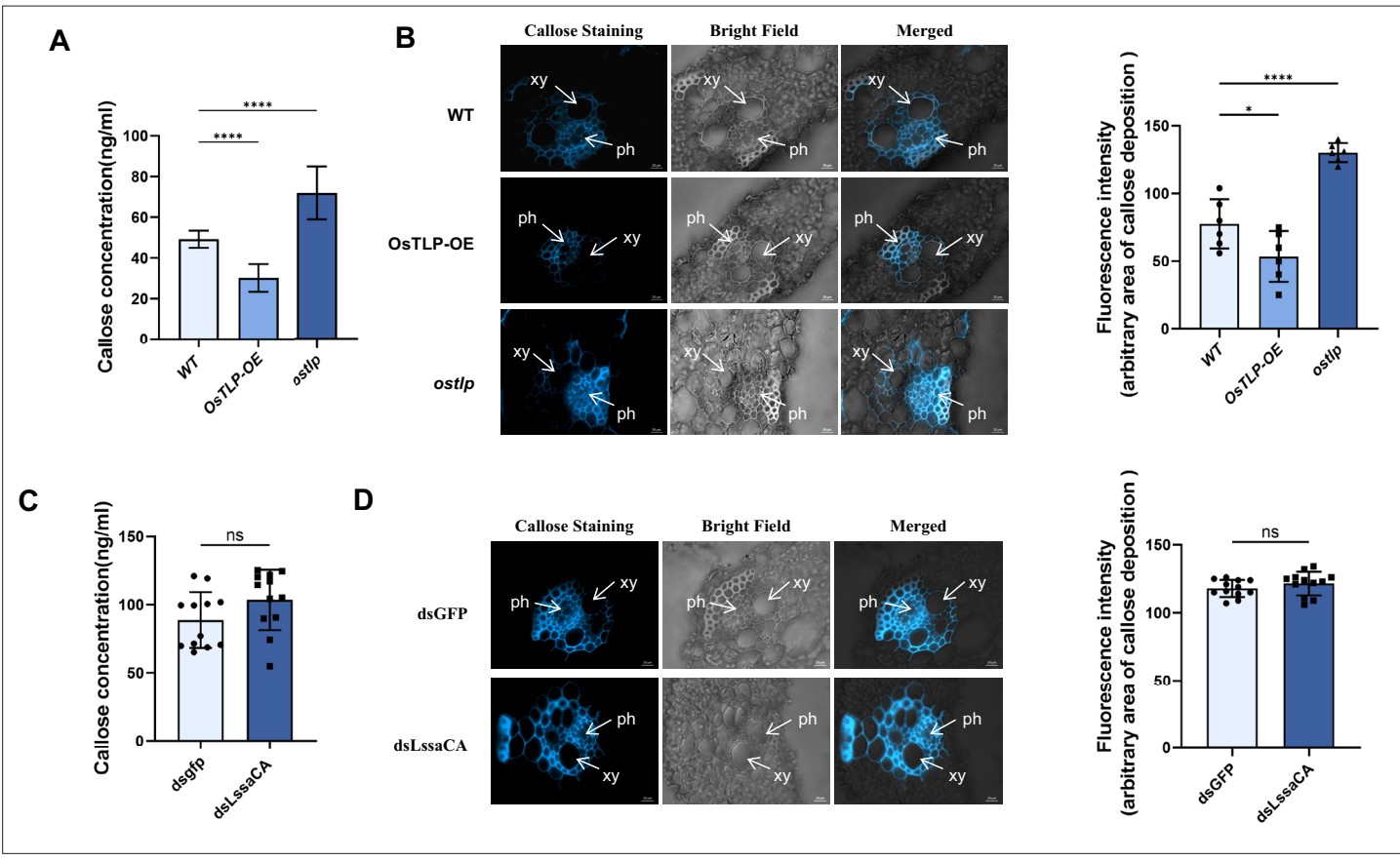

**Figure 6.** *L. striatellus* salivary carbonic anhydrase (LssaCA) mediates in planta callose degradation by enhancing OsTLP enzymatic activity. (**A**) Callose concentration quantified by ELISA in leaves of wild-type (WT), *OsTLP* overexpression (OsTLP OE), and *OsTLP* mutant (*ostlp*) plants. (**B**) Bright blue fluorescence of cross-sections showing callose deposition in leaf phloem. Plants were not fed on by insects. Scale bars: 10 μm. Average fluorescence intensity of callose deposition was measured using ImageJ. Eight to ten random sites per sample were selected for intensity quantification. xy, xylem; ph, phloem. (**C**) Callose concentration quantified by ELISA in leaves of *ostlp* plants fed on by dsGFP- or dsLssaCA-treated RSV-infected *L. striatellus*. (**D**) Bright blue fluorescence of cross-sections showing callose deposition at feeding sites. Samples were prepared from the leaf phloem of *ostlp* plants fed on by dsGFP- or dsLssaCA-treated RSV-infected *L. striatellus*. Scale bars: 10 μm. Average fluorescence intensity was quantified using ImageJ. Eight to ten random sites per sample were selected for intensity quantification. Mean and SD were calculated from three independent experiments. ns, not significant; *p<0.05. ****p<0.0001.

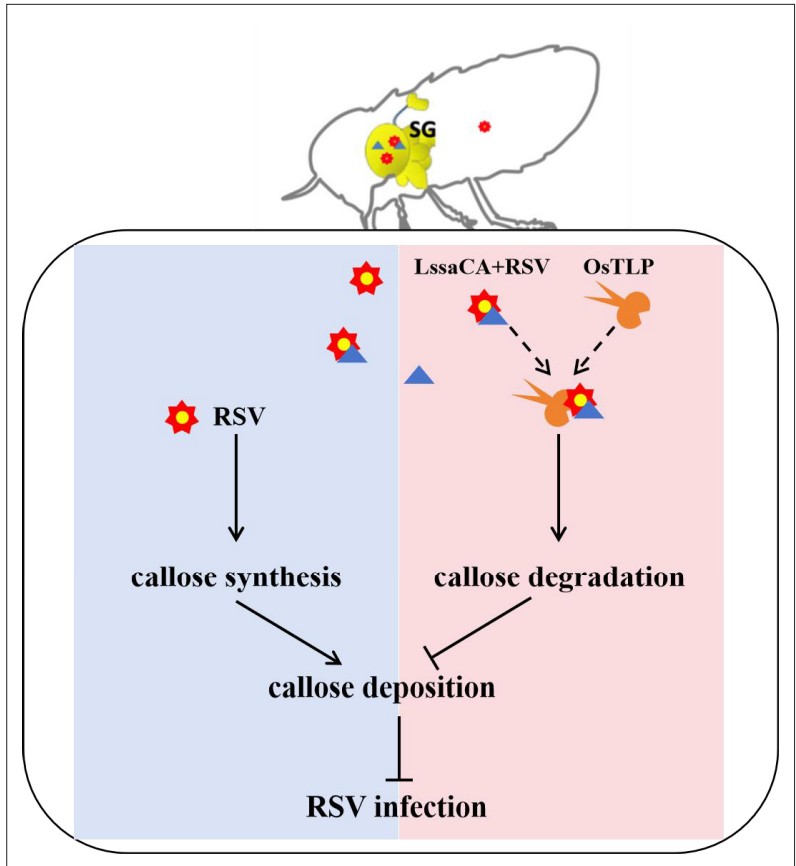

**Figure 7.** Proposed model of rice stripe virus (RSV) NP-LssaCA-OsTLP tripartite interaction facilitating callose degradation and promoting RSV infection. In *L. striatellus* salivary glands (SG), RSV (red star) binds to LssaCA proteins (blue triangle) via the nucleocapsid protein (NP). Upon delivery into the rice phloem, RSV infection triggers callose synthesis, leading to callose deposition that restricts viral spread. However, LssaCA binds to OsTLP (yellow scissors), enhancing its β-1,3-glucanase activity and promoting callose degradation. The RSV NP-LssaCA complex formed in the insect's SG can still interact with OsTLP in planta, resulting in significantly higher OsTLP β-1,3-glucanase activity. This enhanced enzymatic activity efficiently degrades callose, thereby facilitating RSV infection.

Given that LssaCA mediates callose degradation by enhancing OsTLP enzymatic activity, we hypothesized that when OsTLP is knocked out in plant, LssaCA would not affect callose levels induced by RSV infection. To test this, we allowed dsLssaCA-treated viruliferous *L. striatellus* or dsGFP-treated control insects to feed on *ostlp* seedlings. ELISA indicated that LssaCA deficiency did not significantly change callose accumulation (*Figure 6C*). Microscopic analysis confirmed higher callose deposition in phloem cells of *ostlp* seedlings, which remained unaffected by LssaCA deficiency (*Figure 6D*). These results indicate that OsTLP is required in planta for LssaCA-mediated effects on callose deposition.

## Discussion

Insects are known to be vectors of many plant viruses, and some of these plant viruses rely entirely on insect vectors for their plant-to-plant transmission. Insect salivary proteins have been reported to facilitate viral infection of plants through various mechanisms, such as breaking down plant cell walls, inhibiting plant immune defenses and callose deposition, directly binding to viruses to prevent their detection by plant immune systems, and more. In this study, we found that RSV infection of rice plants triggers the deposition of callose. However, this process is reversed by a tripartite interaction involving the central component of an insect salivary protein LssaCA, which subsequently binds to RSV in insect salivary glands and then OsTLP in the plants. The activity of OsTLP's β-1,3-glucanase is

significantly increased when this tripartite interaction occurs, leading to the degradation of callose and thus allowing for efficient RSV infection (*Figure 7*).

Callose is a complex polysaccharide, and its accumulation in plant phloem plays an important role in defending against insect attacks and viral spread (*Zavaliev et al., 2011*). When attacked by piercing insects or viruses, plants activate callose deposition on the sieve plates to occlude the flow of phloem sap, thus discouraging feeding and reducing viral spread. This accumulation of callose in plant phloem can be switched on or off through the action of callose synthases and hydrolases (*Shangguan et al., 2018*). Studies have shown that expression levels of these glucanase enzymes differ between resistant and susceptible plant species. This suggests that some plants may be more successful at resisting infection due to high levels of glucanase enzymes in their tissues, which allow them to more quickly and effectively form callose barriers against insects and viruses (*Hao et al., 2008*). Some insects have evolved β-1,3-glucanase enzymes that can degrade the deposited callose, thus enabling them to bypass the plant's defense mechanisms and continue feeding (*Bucher et al., 2001*; *Hao et al., 2008*). In this study, we demonstrate a novel mechanism by which an insect saliva protein binds to a plant β-1,3-glucanases protein, activating the enzymatic activity and promoting callose degradation.

In this study, we found that plants fed on by viruliferous insects exhibited higher callose deposition than those fed on by uninfected insects (*Figure 5A–C*), suggesting that viral infection *via* insect saliva may constitute a dual stress to the plant, inducing enhanced callose deposition that must be overcome for successful insect feeding and viral spread. To investigate the underlying mechanism, we examined the tripartite interaction between LssaCA-NP-OsTLP. We found that this interaction induced significantly higher β-1,3-glucanase activity than the LssaCA-OsTLP interaction alone, suggesting that SBPH and RSV have evolved mechanisms to enhance β-1,3-glucanase activity in order to bypass the callose-mediated defense deposition and facilitate successful viral spread.

Carbonic anhydrase catalyzes the reversible hydration of carbon dioxide according to the reaction $CO_2 + H_2O \rightleftharpoons HCO_3^- + H^+$. This enzymatic activity is essential for maintaining pH homeostasis in biological systems by regulating the balance between $CO_2$, bicarbonate, and protons. The importance of carbonic anhydrases in pH regulation is exemplified in mosquito larvae, which utilize an extremely alkaline digestive strategy (pH 10.5–11) that is relatively rare in nature, with carbonic anhydrases serving as the central enzymatic system for generating and maintaining bicarbonate/carbonate buffering in the midgut lumen (*Linser et al., 2009*). Similarly, carbonic anhydrases in arthropod saliva can significantly impact host-pathogen interactions through pH modulation. A recent study elucidated how aphid salivary carbonic anhydrase II (CA-II) facilitates virus transmission, with the enzyme showing significantly higher expression levels in winged morphs compared to wingless counterparts. When secreted into the plant apoplastic space, CA-II catalyzes $H^+$ accumulation, resulting in apoplastic acidification from 5.5 to 5.0. This pH reduction enhances polygalacturonase activity, thereby promoting the degradation of demethylesterified homogalacturonan components. Consequently, plants respond by accelerating vesicle trafficking to transport pectin for cell wall reinforcement, a process that simultaneously facilitates viral translocation from the endomembrane system to the apoplastic compartment (*Guo et al., 2023*). In contrast to these pH-dependent mechanisms, our study demonstrated that LssaCA's biological function in mediating RSV infection is, if not completely, at least partially independent of its enzymatic activity. We found that although OsTLP enzymatic activity is highly pH-dependent and exhibits increased enzymatic activity as pH decreases from 7.5 to 5.0 (*Figure 3H*), the LssaCA-OsTLP interaction occurring at pH 7.4 significantly enhanced OsTLP enzymatic activity without any change in buffer pH (*Figure 3G*). These results demonstrate the crucial importance of LssaCA-OsTLP protein interactions, rather than enzymatic activity alone, in mediating RSV infection. While the exact role of LssaCA enzymatic activity in aiding RSV infection remains unclear, the local low-pH plant environment would enhance OsTLP enzymatic activity, suggesting potential synergistic effects between enzymatic and non-enzymatic functions.

In conclusion, this study elucidated a tripartite interaction among a plant virus, an insect saliva protein, and a host phloem β-1,3-glucanase. This interaction inhibited the plant defense of callose deposition and enabled viral phloem transmission. This study provides new insights into the tripartite virus-insect vector-plant interaction, which is relevant to many agriculturally important plant arboviruses whose transmission is facilitated *via* insect saliva proteins.

# Materials and methods

## Viruses, insects, host plants, and antibodies

The RSV-free and RSV-infected *L. striatellus* individuals used in this study were originally captured in Jiangsu Province, China, and were maintained in our laboratory. All plants used to rear *L. striatellus* were grown in a growth incubator (14.5 cm height, 3.2 cm radius) at 25°C under a 16 h light/8 h dark photoperiod. Insects were transferred to fresh seedlings every 15 days to ensure adequate nutrition and insect activity.

RSV ribonucleoproteins (RNPs) were purified from RSV-infected rice plants. These purified RNPs were subsequently used as the antigen to immunize rabbits for the production of an RSV-specific antibody. The LssaCA polyclonal antibody was produced using recombinantly expressed LssaCA protein (with a His-tag at the C-terminus) as the antigen. Anti-His, anti-GST, and anti-MBP antibodies were purchased from ABclonal.

## Tissue collection

Insect dissection and tissue collection were performed using previously described procedures (*Huo et al., 2018*). In brief, insects were anesthetized at 4°C for 10 min, and the forelegs were severed at the coxa-trochanter joint using forceps. The hemolymph was expelled and drawn to the tip of clean forceps. The insects were then dissected from the abdomen in pre-chilled PBS buffer. The fat body was collected with forceps, aided by the tension of the liquid. Tissues, including the guts, salivary glands, and ovaries, were washed three times in PBS to remove contaminants from the hemolymph.

## Bioinformatic analysis

The signal peptide and transmembrane helix were predicted using tools at the SignalP 5.0 server (-5.0) and the TMHMM-2.0 server (-2.0). Homology searches of the protein sequence of LssaCA and OsTLP were conducted using the BLAST algorithm at the National Center for Biotechnology Information (http://blast.ncbi.nlm.nih.gov/Blast.cgi). Conserved domains were predicted with SMART (http://smart.embl-heidelberg.de/).

## Collection of watery and gel saliva

Saliva was collected as described previously (*Huo et al., 2022*). Watery saliva was obtained by centrifuging the collected artificial diet at 10,000×*g* for 10 min to remove the solid gel. To collect gel saliva proteins, the membrane was washed twice with PBS. Sheathes adhering to the parafilm were denatured and solubilized in a solution consisting of 8 M urea, 4% v/v CHAPS, 0.1% w/v sodium dodecyl sulfate (SDS), and 2% v/v DTT. The parafilm was incubated on an orbital shaker at room temperature for 45 min. Then, the denatured gel saliva was concentrated to a final volume of 100 μL. Saliva samples were collected from 2000 insects for silver staining and western blotting analyses.

## Measurement of carbonic anhydrase activity

DNA fragments encoding full-length LssaCA and a LssaCA mutant (H111D, N139H, H141D, H143D, E153H, H166D, and T253E) were amplified by PCR. The resulting products were purified and cloned into the vector pET-30a(+) and pFastBac1 vectors to generate His-tagged recombinant proteins. The pET-30a(+)-LssaCA and pET-30a(+)-LssaCA(mutant) plasmids were transformed into *E. coli* for protein expression. In parallel, the pFastBac1-LssaCA and the pFastBac1-LssaCA(mutant) plasmids were transformed into *E. coli* DH10Bac (Biomed, China) for transposition into the bacmid. Sf9 insect cells were from Thermo Fisher Scientific (USA). According to the supplier's documentation, this cell line was derived from the ATCC CRL-1711 master seed stock and has been tested and confirmed to be free of bacterial, yeast, mycoplasma, and viral contamination. Sf9 cells were cultured in suspension in SIM SF Expression Medium (Sino Biological, China) at 27°C. Prior to transfection, cells were examined under a microscope to confirm they were in the logarithmic growth phase with optimal morphology. Recombinant bacmids were then transfected into *Spodoptera frugiperda* (Sf9) cells using Cellfectin II Reagent (Invitrogen, USA). Recombinant proteins expressed in both systems were purified using Ni Sepharose resin (GE Healthcare, Uppsala, Sweden).

Carbonic anhydrase activity was assessed by measuring esterase activity using *p*-nitroethyl acetate (*p*-NPA) as the substrate. LssaCA and its mutant were expressed in *E. coli* and Sf9 cells with C-terminal

His-tag, and the purified proteins were diluted in PBS buffer (pH 7.0) to a final concentration of 1 mg/mL. Reaction conditions followed the method described by *Verpoorte et al., 1967*, with minor modifications. Briefly, 1 mL of freshly prepared 3 mM *p*-NPA (dissolved in acetone) was added to 1.9 mL of 100 mM PBS (pH 7.0), followed by 0.1 mL of LssaCA enzyme solution (0.5 mg/mL in 100 mM PBS). Absorbance at 384 nm was measured at the start of the reaction and after 3 minutes. One unit (U) of enzyme activity was defined as the amount of enzyme required to hydrolyze 1 μmol of *p*-NPA per minute at pH 7.0 and 25°C. A blank control consisted of 2 mL of 100 mM PBS (pH 7.0) mixed with 1 mL of 3 mM *p*-NPA.

## Immunofluorescence assays

Immunofluorescence assays (IFA) were performed to determine the localization of RSV and LssaCA in *L. striatellus* salivary glands. The dissected salivary gland samples were fixed in 4% v/v paraformaldehyde at 4 °C overnight and then permeabilized in 2% v/v Triton X-100 for 2 h. The samples were incubated with rabbit anti-RSV serum and mouse anti-LssaCA serum (1:250 dilution in PBST/FBS: PBS containing 2% v/v Triton X-100 and 2% fetal bovine serum) for 1 h at room temperature. After washing twice with PBST, the samples were incubated with 1:200 diluted Alexa Fluor 488-conjugated goat anti-mouse antibody and Alexa Fluor 594-conjugated goat anti-rabbit antibody (Invitrogen) for 1 h. After washing three times with PBST, the samples were examined under a Leica TCS SP8 microscope (Leica, Wetzlar, Germany) with the following settings: (1) Alexa Fluor 488 channel: excitation at 488 nm using an argon laser; emission detected with a 493–553 nm bandpass filter. Alexa Fluor 594 channel: excitation at 594 nm using a HeNe laser; excitation detected with a 600–650 nm bandpass filter.

## Electrical penetration graph (EPG)

EPG was performed according to previously reported (*Huo et al., 2022*). *L. striatellus* individuals were pre-starved for 4 h and then anesthetized for 10 min at 4°C. A gold wire (20 μm diameter, 4 cm long, connected to the EPG probe via a copper nail) was attached to the dorsal thorax of the insect using water-soluble silver glue. When the glue had dried, the probe was connected to the amplifier and the insect was placed on the leaf of a 2-week-old rice plant in a pot. The other amplifier was connected to the soil in the pot with another brass wire. The electronic signals were amplified, and the digitized data were received by a computer using Style software (Wageningen University, Wageningen, Netherlands). Data were recorded for each insect continuously for 12 h.

## Evaluation of callose deposition in rice plants

Five insects were allowed to feed for 24 h on a 2×2 cm leaf area of 10-d-old rice plants. After removal of the insects, leaf tissues were collected from both the feeding site and an adjacent undamaged area located 1 cm away (non-fed site). Samples were immersed in 5 mL alcoholic lactophenol solution (1 volume of phenol:glycerol:lactic acid:water in a 1:1:1:1 ratio, mixed with 2 volumes of ethanol), evacuated with a syringe for 15 min, and then incubated at 65°C for 30 min until the chlorophyll was completely cleared. Cleared leaf samples were rinsed with 50% v/v ethanol, then with water, and then stained in 150 mM $K_2HPO_4$ (pH 9.5) containing 0.01% w/v aniline blue for 30 min in the dark. Stained leaf samples were sectioned and mounted in 50% v/v glycerol on slides. The slides were examined under a Zeiss observer Z1 immunofluorescence microscope (Carl Zeiss MicroImaging GmbH, Göttingen, Germany) with excitation at 405 nm. Fluorescence intensity was quantified using ImageJ software. For ImageJ analysis, images were converted to 8-bit grayscale and adjusted for brightness. Phloem-specific regions of interest were outlined using the freehand selection tools. A standardized fluorescence threshold (set at >2 × the mean background intensity) was applied to isolate callose signals. Integrated density and area values were used to calculate normalized fluorescence intensity. Identical exposure times were maintained across all independent experiments.

Callose concentrations were quantified using a plant callose ELISA kit (Catalog No. JL46407; Jonln, Shanghai, China), following the manufacturer's instruction. Leaf tissues from feeding and non-fed sites were homogenized in PBS buffer (pH 7.4), and the homogenate was centrifuged at 500–1000 × *g* for 20 min. The supernatant was collected, and absorbance at 450 nm was measured. Callose concentrations were calculated using a standard curve.

## RNA interference

We conducted RNAi analyses to determine the function of LssaCA in mediating RSV infection and regulating callose deposition. The *LssaCA*-specific gene fragment was PCR-amplified with the primer pair LssaCA-T7F/LssaCA-T7R (*Supplementary file 4*). Then, dsRNA was synthesized using a commercial kit (T7 RiboMAX Express RNAi System, Promega, Madison, WI, USA) and purified by phenol: chloroform extraction and isopropanol precipitation. Finally, 36.8 nL dsRNA at 1 ng/nL was delivered into the insect hemocoel for gene silencing. Microinjection was performed using a glass needle through a Nanoliter 2000 (World Precision Instruments, Sarasota, FL, USA). The negative control, GFP dsRNA, was synthesized and microinjected following the same protocol.

## RT-qPCR

Gene transcript levels were determined by RT-qPCR. The RNA was extracted from *L. striatellus* tissues using a RNeasy Mini Kit (QIAGEN, Hilden, Germany), and from plant leaves using TRIzol. Reverse transcriptional PCR (iScript cDNA Synthesis Kit, Bio-Rad, Hercules, CA, USA) and SYBR-Green-based qPCR (SYBR Green Supermix, Bio-Rad) was performed according to the manufacturer's protocols. The primer pair used to amplify *LssaCA* was LssaCA-q-F/LssaCA-q-R (*Supplementary file 4*). Viral RNA copies were measured using the primers NP-q-F/NP-q-R (*Supplementary file 4*). *L. striatellus* translation elongation factor 2 (*EF2*) and rice *actin* were amplified as internal controls to ensure equal loading of cDNA isolated from different samples. The primer pairs EF2-q-F/EF2-q-R and actin-q-F/actin-q-R (*Supplementary file 4*) were used to amplify *EF2* and *actin*, respectively. Water was used as a negative control.

## Yeast two-hybrid assays

Yeast two-hybrid screening was performed to identify rice proteins interacting with LssaCA. A high-complexity rice cDNA library was fused to *Gal4 AD* and transformed into the yeast strain Y187. The *LssaCA* gene lacking the signal-peptide sequence (*LssaCA^sp-^*) was fused to *Gal4 DNA-BD* and transformed into the yeast strain Y2HGold. The two yeast strains were co-cultured overnight to allow mating to create diploids. After library screening, positive clones were selected on quadruple dropout medium: SD/–Ade/–His/–Leu/–Trp supplemented with X-a-Gal and aureobasidin A (QDO/X/A), and prey plasmids were isolated for sequencing. To further confirm the interaction between LssaCA and OsTLP, *LssaCA^sp-^* was cloned into the prey vector pDB-Leu, and the *OsTLP* gene lacking the signal-peptide sequence (OsTLP^sp-^) was cloned into the bait vector pPC86. The two plasmids were co-transformed into the yeast strain Mav203, and the transformants were selected on SD/–His–Leu–Trp +20 mM 3-AT medium.

## Pull-down assays

Pull-down assays were performed to determine the molecular interactions between LssaCA and RSV NP, LssaCA and OsTLP, and OsTLP and the LssaCA-NP complex. The LssaCA sequence lacking the signal peptide (*LssaCA^sp-^*) was cloned into the pGEX-3x and pET-30a(+) vectors to generate GST-tagged (GST-LssaCA) and C-terminal His-tagged (LssaCA-His) fusion proteins, respectively. Similarly, the full-length NP gene was cloned into the same vectors to produce GST-NP and NP-His. The OsTLP sequence without its signal peptide (*OsTLP^sp-^*) was cloned into the pMAL-p2X vector to express an N-terminal MBP-fusion protein (MBP-OsTLP). All recombinant proteins were expressed in and purified from *E. coli* strain BL21 for subsequent assays.

The interaction between LssaCA and RSV NP was determined using GST pull-down analyses as described previously (*Huo et al., 2014*). Briefly, GST was used as a negative control. GST-LssaCA or GST protein was bound to Glutathione Sepharose beads (Cytiva, Marlborough, MA, USA) for 2 h at 4°C. Subsequently, purified NP-His protein was added to the beads and incubated for another 2 h at 4°C. After centrifugation and five washes, the bead-bound proteins were separated by SDS-PAGE and detected by western blotting with antibodies against GST (Tiangen, Beijing, China) or His-tag (Tiangen).

Next, MBP pull-down analyses were performed to determine the interaction involving OsTLP, LssaCA, and RSV NP. MBP was used as a negative control. The purified MBP-OsTLP or MBP protein was bound to amylose resin (New England Biolabs, Beverley, MA, USA) for 2 h at 4°C, and then the LssaCA-His protein was added to the beads. To determine the interaction between OsTLP and the

LssaCA-NP complex, a separate reaction was performed in which LssaCA-His and GST-NP proteins were pre-incubated together for 2 h at 4°C to form the complex before addition to the resin. An equivalent amount of LssaCA-His was used in both reaction systems. After centrifugation and five washes, the bead-bound proteins were separated by SDS-PAGE and detected by western blotting with the MBP antibody (Cell Signaling Technology, Danvers, MA, USA), His-tag antibody (Tiangen) and GST antibody (Tiangen).

## Microscale thermophoresis (MST) assays

We conducted MST assays to detect the interactions between LssaCA and NP, and between LssaCA and OsTLP. First, 10 µM purified LssaCA$^{sp-}$ protein (His-fusion protein) was labeled with a Monolith NT Protein Labeling Kit RED-NHS (Nano Temper Technologies GMBH, München, Germany) using red fluorescent dye NT-647 N-hydroxysuccinimide (amine-reactive), according to the manufacturer's instructions. The binding assays were performed on a Monolith NT.115 Microscale Thermophoresis instrument (Nano Temper Technologies GMBH) using capillaries treated using the standard method. The labeled protein LssaCA$^{sp-}$ was added to serially diluted NP (GST-NP, GST was used as a negative control) or OsTLP$^{sp-}$ (MBP-fusion protein, MBP was used as a negative control). The initial concentration of each protein was 40 µM with 0.1% (v/v) Tween 20. The KD Fit function of NanoTemper Analysis software version 1.2.214.1 was used for curve fitting and to calculate the dissociation constant (Kd).

## β-1,3-Glucanase activity assays

β-1,3-glucanase hydrolyzes laminarin by cleaving β-1,3-glucoside bond, thereby generating sugar termini. Enzyme activity was quantified by measuring the rate of reducing sugars production. One unit (U) of β-1,3-glucanase activity was defined as the amount of enzyme that hydrolyzes laminarin to generate 1 µg reducing sugars per minute.

The β-1,3-glucanase activity of OsTLP was measured using a beta-1,3-glucanase microplate assay kit (Catalog No. ASK1028, Bioworld, Bloomington, MN, USA) according to the manufacturer's instructions. To determine the enzymatic activity of recombinant OsTLP proteins, MBP-fused OsTLP$^{sp-}$ protein and MBP control protein were adjusted to 0.5 mg/mL. The MBP-OsTLP$^{sp-}$ fusion protein was equilibrated in buffers of varying pH for the assay. To determine the enzymatic activity of OsTLP expressed in planta, 0.1 g of rice leaf tissue from either OsTLP over-expressing or wild-type plants was homogenized with 1 mL of assay buffer on ice. The homogenate was centrifuged at 12,000 × $g$ at 4°C for 10 min, and the resulting supernatant was transferred into a new tube and kept on ice until analysis.

For the assay, 50 µL of the protein solution was mixed with laminarin substrate and incubated at 37°C for 30 min. The reaction was terminated by incubating the reaction tube in a boiling water bath for 10 min. The reaction supernatant was transferred into a microplate and incubated at 90°C for 10 min before measuring the absorbance at 540 nm.

## Generation of transgenic plants

The full-length *OsTLP* gene was PCR-amplified with the primer pair TLP-F/TLP-R (*Supplementary file 4*) and cloned into the plant expression vector pCAMBIA1300 to produce the plasmid pCAMBIA1300-OsTLP. This construct was transformed into seedlings of the *Japonica* rice cultivar Nipponbare through *Agrobacterium*-mediated transformation, following the protocol described by *Hiei et al., 1994*. The *ostlp* knockout line was generated using CRISPR-Cas9 technology according to established protocols. Thirty independent T$_0$ transformants were obtained and screened by germinating seeds from T$_2$ lines on medium containing 50 µg/ml hygromycin. Hygromycin-resistant homozygous lines were selected for further experiments. Knockout events were confirmed by sequencing with specific primers (*Supplementary file 3*). Seeds from homozygous lines transformed with the vector pCAMBIA1300-eGFP were used as negative control.

## Statistical analyses

Graphs were generated, and statistical analyses were conducted using Prism 9.0 software (GraphPad Software, San Diego, CA, USA). Data are presented as mean ± standard deviation (SD) from two or three independent experiments. Statistical significance between two groups was assessed using an unpaired Student's *t*-test, while comparisons among three or more groups were performed using one-way analysis of variance.

## Acknowledgements

This work was supported by the National Key R&D Program of China (2022YFD1400800), Major Program of the National Natural Science Foundation of China (32090013), and Youth Innovation Promotion Association CAS (2021084).

## Additional information

### Funding

| Funder | Grant reference number | Author |
|---|---|---|
| National Key Research and Development Program of China | 2022YFD1400800 | Lili Zhang |
| National Natural Science Foundation of China | 32090013 | Lili Zhang |
| Youth Innovation Promotion Association of the Chinese Academy of Sciences | 2021084 | Yan Huo |

The funders had no role in study design, data collection and interpretation, or the decision to submit the work for publication.

### Author contributions

Jing Zhao, Conceptualization, Investigation, Methodology, Writing – original draft; Xiangyi Meng, Jie Yang, Investigation, Methodology; Rongxiang Fang, Supervision; Yan Huo, Conceptualization, Funding acquisition, Investigation, Methodology, Writing – original draft, Writing – review and editing; Lili Zhang, Conceptualization, Supervision, Funding acquisition, Writing – original draft, Writing – review and editing

### Author ORCIDs

Yan Huo  https://orcid.org/0000-0002-5783-1708
Lili Zhang  https://orcid.org/0000-0002-4321-0521

Reviewer #2 (Public Review): https://doi.org/10.7554/eLife.88132.4.sa1
Author response https://doi.org/10.7554/eLife.88132.4.sa2

## Additional files

### Supplementary files

Supplementary file 1. Summary of amino acid substitutions in the LssaCA mutant.

Supplementary file 2. Yeast two-hybrid screening for rice proteins interacting with LssaCA.

Supplementary file 3. The sgRNA target site.

Supplementary file 4. List of primers used in this study.

MDAR checklist

### Data availability

All data is available at https://doi.org/10.5061/dryad.jh9w0vtqz.

The following dataset was generated:

| Author(s) | Year | Dataset title | Dataset URL | Database and Identifier |
|---|---|---|---|---|
| Huo Y, Meng XY, Zhao J, Zhang LL | 2025 | Rice stripe virus utilizes a Laodelphax striatellus salivary carbonic anhydrase to facilitate plant infection by direct molecular interaction | https://doi.org/10.5061/dryad.jh9w0vtqz | Dryad Digital Repository, 10.5061/dryad.jh9w0vtqz |

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
