## [Editor Report · eLife Assessment]

This **valuable** study presents a well-designed set of experiments demonstrating how a planthopper salivary carbonic anhydrase can promote rice stripe virus infection by modulating callose deposition in the host plant. The authors provide **solid** data for the proposed protein–protein interactions, including strengthened evidence for the LssaCA-NP-OsTLP complex and clarified dynamics of LssaCA presence in planta. Overall, the work reveals a mechanistic link whereby a vector salivary protein enhances a plant β-1,3-glucanase to suppress callose-based defense, thereby facilitating early viral establishment.

---

## [Referee Report · Reviewer #2 (Public Review)]

There is increasing evidence that viruses manipulate vectors and hosts to facilitate transmission. For arthropods, saliva plays an essential role for successful feeding on a host and consequently for arthropod-borne viruses that are transmitted during arthropod feeding on new hosts. This is so because saliva constitutes the interaction interface between arthropod and host and contains many enzymes and effectors that allow feeding on a compatible host by neutralizing host defenses. Therefore, it is not surprising that viruses change saliva composition or use saliva proteins to provoke altered vector-host interactions that are favorable for virus transmission. However, detailed mechanistic analyses are scarce. Here, Zhao and coworkers study transmission of rice stripe virus (RSV) by the planthopper Laodelphax striatellus. RSV infects plants as well as the vector, accumulates in salivary glands and is injected together with saliva into a new host during vector feeding.

The authors present evidence that a saliva-contained enzyme - carbonic anhydrase (CA) - might facilitate virus infection of rice by interfering with callose deposition, a plant defense response. In vitro pull-down experiments, yeast two hybrid assay and binding affinity assays show convincingly interaction between CA and a plant thaumatin-like protein (TLP) that degrades callose. Similar experiments show that CA and TLP interact with the RSV nuclear capsid protein NT to form a complex. Formation of the CA-TLP complex increases TLP activity by roughly 30% and integration of NT increases TLP activity further. This correlates with lower callose content in RSV-infected plants and higher virus titer. Further, silencing CA in vectors decreases virus titers in infected plants. Interestingly, aphid CA was found to play a role in plant infection with two non-persistent non-circulative viruses, turnip mosaic virus and cucumber mosaic virus (Guo et al. 2023 doi.org/10.1073/pnas.2222040120), but the proposed mode of action is entirely different.

Editors' note: this version was assessed by the editors, without further input from the reviewers.

---

## [Author Response]

The following is the authors’ response to the previous reviews

**Reviewer #1 (Public review):**
In this study, the authors identified an insect salivary protein LssaCA participating viral initial infection in plant host. LssaCA directly bond to RSV nucleocapsid protein and then interacted with a rice OsTLP that possessed endo-β-1,3-glucanase activity to enhance OsTLP enzymatic activity and degrade callose caused by insects feeding. The manuscript suffers from fundamental logical issues, making its central narrative highly unconvincing.(1) These results suggested that LssaCA promoted RSV infection through a mechanism occurring not in insects or during early stages of viral entry in plants, but in planta after viral inoculation. As we all know that callose deposition affects the feeding of piercing-sucking insects and viral entry, this is contradictory to the results in Fig. S4 and Fig. 2. It is difficult to understand callose functioned in virus reproduction in 3 days post virus inoculation. And authors also avoided to explain this mechanism.

We appreciate your insightful comment and acknowledge that our initial description may not have been sufficiently clear.

(1) Based on the EPG results, we found that LssaCA deficiency did not significantly affect total feeding time, time to first non-phloem phase, or time to first phloem feeding (Fig. S8A-D in the revised manuscript). However, the continuity of sap ingestion was disturbed—the N4 waveform of dsLssaCA SBPHs was occasionally interrupted for brief periods (newly added Fig. S8E in the revised manuscript), likely due to phloem blockage. In the revised manuscript, we have added this analysis to the Result section (Lines 285-291 and 578-587) and provided the EPG procedure in Material and Methods section (Lines 670-680).

(2) We assessed RSV titers immediately post-feeding to confirm the inoculation viral loads (Fig. 2G) and at 3 dpf (Fig. 2H-I) to assess the in-planta effects following viral inoculation. This did not mean that callose functions in virus reproduction at 3 days post viral inoculation. Rather, callose deposition typically occurs immediately in response to insect feeding and virus inoculation. When measuring callose deposition, we allowed insects to feed for 24 h and quantified the callose levels immediately post feeding. The EPG results showed that sap ingestion continuity was disrupted—the N4 waveform of dsLssaCA-treated SBPHs was occasionally interrupted for brief periods (newly added Fig. S8E in the revised manuscript), likely due to phloem blockage. We have reorganized the description to avoid confusion. Please see Lines 139-144 and Fig. S8E for detail.

(1) Missing significant data. For example, the phenotypes of the transgenic plants, the RSV titers in the transgenic plants (OsTLP OE, ostlp). The staining of callose deposition were also hard to convince. The evidence about RSV NP-LssaCA-OsTLP tripartite interaction to enhance OsTLP enzymatic activity is not enough.

We thank the reviewer for this insightful comment.

(1) We constructed OsTLP overexpression and mutant transgenic plants (OsTLP OE and ostlp) and assessed their phenotypes regarding RSV infection levels. Compared with wild-type plants, OsTLP OE plants exhibited accelerated growth, while ostlp plants showed growth inhibition. Following feeding by viruliferous L. striatellus, OsTLP OE plants had significantly higher RSV titers compared with wild-type plants, whereas ostlp mutant plants exhibited significantly lower RSV titers (Lines 221-228 and new Fig. 3I). These results indicate that OsTLP facilitates RSV infection in planta.

(2) The images showing callose deposition staining are representative of 15 images from 3 independent insect treatments. In addition to the staining images, we quantified fluorescence intensity and measured callose concentration by ELISA.

(2) Figure 4a, there was the LssaCA signal in the fourth lane of pull-down data. Did MBP also bind LsssCA? The characterization of pull-down methods was rough a little bit. The method of GST pull-down and MBP pull-down should be characterized more in more detail.

We thank the reviewer for this helpful comment. MBP did not bind LssaCA. We have repeated the pull-down experiment and provide clearer figure with improved results. We have also revised and provided more detailed descriptions of the GST pull-down and MBP pull-down methods. Please refer to Lines 744-774 and Figure 4A for details.